# Can *Acropora tenuis* larvae attract native Symbiodiniaceae cells by green fluorescence at the initial establishment of symbiosis?

**Hiroshi Yamashita**[1]*, **Kazuhiko Koike**[2], **Chuya Shinzato**[3], **Mitsuru Jimbo**[4], **Go Suzuki**[1]

**1** Fisheries Technology Institute, Japan Fisheries Research and Education Agency, Ishigaki, Okinawa, Japan, **2** Graduate School of Integrated Sciences for Life, Hiroshima University, Higashi-Hiroshima, Hiroshima, Japan, **3** Atmosphere and Ocean Research Institute, The University of Tokyo, Kashiwa, Chiba, Japan, **4** School of Marine Biosciences, Kitasato University, Sagamihara, Kanagawa, Japan

* hyamashita@fra.affrc.go.jp

**Data Availability Statement:** All relevant data are within the manuscript and its Supporting Information files.

## Abstract

Most corals acquire symbiodiniacean symbionts from the surrounding environment to initiate symbiosis. The cell densities of Symbiodiniaceae in the environment are usually low, and mechanisms may exist by which new coral generations attract suitable endosymbionts. Phototaxis of suitable symbiodiniacean cells toward green fluorescence in corals has been proposed as one such mechanism. In the present study, we observed the phototaxis action wavelength of various strains of Symbiodiniaceae and the fluorescence spectra of aposymbiotic *Acropora tenuis* larvae at the time of endosymbiont uptake. The phototaxis patterns varied among the Symbiodiniaceae species and "native" endosymbionts—commonly found in *Acropora* juveniles present in natural environments; that is, *Symbiodinium microadriaticum* was attracted to blue light rather than to green light. Another native endosymbiont, *Durusdinium trenchii*, showed no phototaxis specific to any wavelength. Although the larvae exhibited green and broad orange fluorescence under blue-violet excitation light, the maximum green fluorescence peak did not coincide with that of the phototaxis action spectrum of *S. microadriaticum*. Rather, around the peak wavelength of larval green fluorescence, this native endosymbiont showed slightly negative phototaxis, suggesting that the green fluorescence of *A. tenuis* larvae may not play a role in the initial attraction of native endosymbionts. Conversely, broad blue larval fluorescence under UV-A excitation covered the maximum phototaxis action wavelength of *S. microadriaticum*. We also conducted infection tests using native endosymbionts and aposymbiotic larvae under red LED light that does not excite visible larval fluorescence. Almost all larvae failed to acquire *S. microadriaticum* cells, whereas *D. trenchii* cells were acquired by larvae even under red illumination. Thus, attraction mechanisms other than visible fluorescence might exist, at least in the case of *D. trenchii*. Our results suggest that further investigation and discussion, not limited to green fluorescence, would be required to elucidate the initial attraction mechanisms.

**Funding:** This work was supported by Japan Society for the Promotion of Science (JSPS) Grants-in-Aid for Scientific Research (KAKENHI) Grant Numbers 26291094 and 18H02270 for HY., 17KT0027 and 20H03235 for CS., 24570028 for KK. The funders had no role in study design, data collection and analysis, decision to publish, or preparation of the manuscript.

**Competing interests:** The authors have declared that no competing interests exist.

## Introduction

Members of the dinoflagellate family Symbiodiniaceae, often called the zooxanthellae, are well-known coral endosymbionts. Corals require photosynthetic products of symbiodiniacean cells (e.g.,[1]), and endosymbionts are essential for coral survival. However, the early life stages of the majority of broadcast-spawning corals, including *Acropora* corals, lack symbiodiniacean symbionts [2]. These corals must acquire endosymbiont cells from the surrounding environment to establish a symbiotic relationship.

The family Symbiodiniaceae includes ten genetic groups referred to as clades (A–J: [3]), and recently some clades or distinct sub-clades have been reclassified as genera [4, 5]. Furthermore, most of the clade or genus consists of numerous "types" or species (e.g., [6]). The Symbiodiniaceae are highly diverse, and all of the genera/clades are found in Pacific coral reef environments (i.e., water column or sediments) [3, 7–12]. Thus, new generations of corals are likely to acquire these diverse environmental symbiodiniaceans. However, natural *Acropora* coral juveniles often harbor specific genera of Symbiodiniaceae; namely, genus *Symbiodinium* (former clade A) and genus *Durusdinium* (former clade D) [10, 13, 14]. Environmental symbiodiniacean cell densities are typically low, and these native endosymbiont candidates are especially rare [10]. Thus, some mechanisms probably exist to allow aposymbiotic corals to encounter suitable endosymbionts. Laboratory experiments have shown that some cultured Symbiodiniaceae strains are attracted to aposymbiotic *Acropora tenuis* larvae [14]. Symbiodiniaceae cells display chemosensory responses [15–17] and exhibit phototaxis toward green light [18]. These properties may assist encounters between symbiodiniacean cells and aposymbiotic coral larvae or juveniles. For example, Pasternak et al. [19] demonstrated that symbiodiniacean cells are attracted to exudate-solution drops of aposymbiotic juvenile polyps of the soft coral *Heteroxenia fuscescens*. Aihara et al. [20] also demonstrated that green fluorescent protein (GFP) can attract a cultured strain of Symbiodiniaceae, using the green morph *Echinophyllia aspera* and a green fluorescent dye. Coral larvae display a GFP-like fluorescence [21]. Hollingsworth et al. [18, 22] proposed a beacon hypothesis, suggesting that GFP might play a role in attracting symbiodiniacean cells to aposymbiotic larvae. Symbiodiniacean cells in the motile phase possess eye spots composed of crystalline layers that can refract/polarize light and deflect UV/blue/green wavelength light [23]. Therefore, the beacon hypothesis is a plausible scenario for the initial encounter of early life stage corals and their endosymbiont candidates. However, GFP is not only a characteristic of aposymbiotic larvae/juveniles, but also of adult corals. If endosymbiont candidates are attracted to coral fluorescence, the question remains as to how they reach tiny larvae/juveniles rather than large adult corals. Although a previous study reported that some strains of Symbiodiniaceae can be attracted to green light and green fluorescence [18, 20], it is unclear whether species of Symbiodiniaceae that are native to *Acropora* juveniles are attracted to green fluorescence. Further, the fluorescence spectra of larvae at the time of endosymbiont uptake remain unknown. This lack of information obscures the mechanism of Symbiodiniaceae cell attraction by aposymbiotic corals. In the present study, to clarify this mechanism, we observed the fluorescence spectra of aposymbiotic *A. tenuis* larvae and phototaxis of several cultured Symbiodiniaceae strains, including native endosymbionts of natural *Acropora* juveniles.

## Materials and methods

### Phototaxis of Symbiodiniaceae cells

**Cultured strains.** Eight species comprising six genera of Symbiodiniaceae were used for phototaxis measurements. These included AJIS2-C2 (*Symbiodinium microadriaticum*, former

clade A type A1), GTP-A6-Sy (*S. natans*, former clade A type A2-relative), CS-161 (*S. tridacnidorum*, former clade A type A3), CCMP1633 (*Breviolum* sp. former clade B), CCMP2466 (*Cladocopium goreaui*, former clade C type C1), CCMP2556 (*Durusdinium trenchii*, former clade D type D1-4), MJa-B6-Sy (*Effrenium voratum*, former clade E), and CS-156 (*Fugacium kawagutii*, former clade F). CS-161 and CS-156 were purchased from the Commonwealth Scientific & Industrial Research Organization (ACT, Australia), and CCMP1633, CCMP2466, and CCMP2556 were purchased from Provasoli–Guillard National Center for Culture of Marine Algae and Microbiota (ME, USA). The other strains were originally isolated by Yamashita and Koike [9]. In the wild, *Acropora* coral juveniles harbor *S. microadriaticum*, *S. tridacnidorum*, *C. goreaui*, and *D. trenchii* e.g., [10, 14]; among these, *S. microadriaticum* and *D. trenchii* are major endosymbionts, and cultured strains of these two species (AJIS2-C2 and CCMP2556) are easily acquired by *A. tenuis* larvae even in laboratory experiments [14]. Symbiodiniaceae cultured strains were maintained in an incubator at 27˚C under a light regime of 80–120 μmol photon m$^{-2}$ s$^{-1}$ (12:12 h [light:dark] period) in IMK medium (Sanko Jyunyaku, Tokyo, Japan).

**Phototaxis measurements.** The cultured strains were inoculated into IMK medium-filled 250-mL cell culture flasks wrapped with black PVC tape to create grid openings (Fig 1A) and incubated in the aforementioned incubator for several days. During the early to mid-logarithmic proliferation phase of each strain, band-pass filters (ø 25 mm) were attached to the grid area on the cell culture flasks with double-sided tape (Fig 1B). We used eight types of band-pass optical filters: 330–385 nm, 400–410 nm, central wavelength (CWL) at 470 nm, 490 nm, 510 nm, 530 nm, 550 nm, and 570 nm. Band-pass filters for 330–385 nm and 400–410 nm were originally excitation filters of fluorescence filter cubes of units "U-MWU" and "U-MNV" (Olympus, Tokyo, Japan), respectively. Other filters were purchased from Thorlabs Japan Inc. (Tokyo, Japan), and full width at half maximum (FWHM) was 10 nm. The cell culture flask, to which these optical filters were attached in a random order, was then placed in front of a

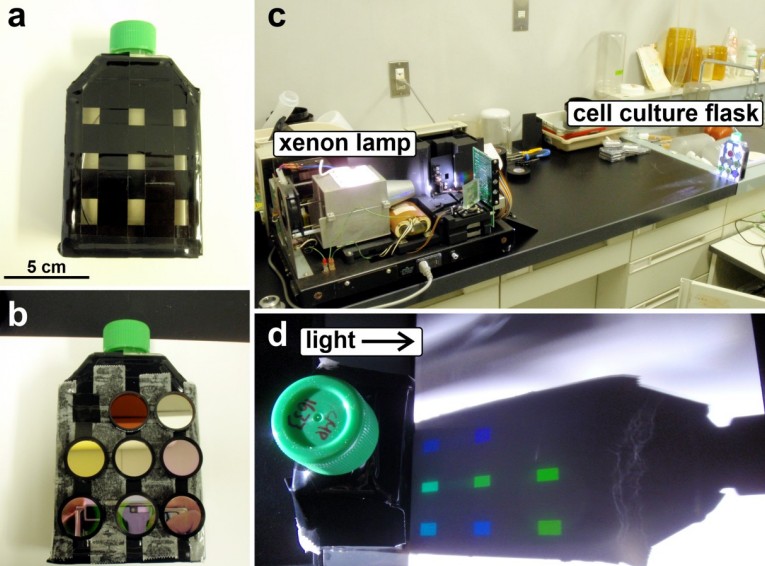

**Fig 1. A cell culture flask and xenon lamp were used to assess the phototaxis of Symbiodiniaceae cultured strains.** a) 250-mL cell culture flask with black PVC tape attached in a grid shape. b) Eight types of band-pass optical filters (ø 25 mm) were attached with double-sided tape. c) The cell culture flask with band-pass filters was placed in front of the xenon lamp source. d) A specific wavelength of light passed through each band-pass filter into the cell culture flask. To count the symbiodiniacean cells, 1 mL of medium was collected from each grid using a Pasteur pipette inserted from the mouth of cell culture flask.

xenon lamp source (Fig 1C). We took 1 mL of medium from a gently but well-mixed cell culture flask before the phototaxis measurements, to confirm initial cell density. After turning on the xenon lamp for ten min, 1 mL of the medium was carefully collected using a Pasteur pipette near the grids attached to each optical filter, starting at the top line (Fig 1D) to minimize disturbance. The cells were then counted under a microscope. Cells were counted three times for 2 μL of each 1 mL sample collected, and cell numbers were averaged. The positive/negative phototaxis for each wavelength was calculated as follows:

(average cell numbers after lamp irradiation) / (average initial cell numbers) × 100

This ratio reflects the increase/decrease in cell numbers near the grids attached to each wavelength band-pass filter. These measurements were performed three times with randomly reattached filters (S1 Dataset). All experiments were performed in the morning because motile cell ratios are high in the morning for all experimental cultured strains [24].

## Fluorescence of *Acropora tenuis* larvae

**Preparations of *A. tenuis* larvae.** Egg/sperm bundles were collected from nine colonies of *A. tenuis* kept in running natural seawater tanks. Parental *A. tenuis* colonies were collected from Sekisei lagoon between Ishigaki and Iriomote island in the southern part of Okinawa, Japan. Sampling of corals was permitted by the Okinawa Prefectural Government for research use (No. 29–74). Fertilized eggs were washed to remove any remaining sperm and unexpected symbiodiniacean cell contaminants using ultrafiltered (UF) seawater (Membrane Technology Co., Okinawa, Japan) and then kept in the incubator, as described above. The water was changed once a day, using UF seawater.

**Fluorescent micrographs and measurements of fluorescence of *A. tenuis* larvae.** Larval fluorescence was observed at larval ages of 1, 2, 3, 4, 5, 6, 7, 8, 9, 12, 16, and 21 d under an epifluorescence microscope (BX50, 20× objective lens, Olympus, Tokyo, Japan). Fluorescence filter cubes were U-MWBV2 (Ex. 400–440 nm band-pass, Em. ≥475 nm longpass, Dichroic Mirror 455 nm; blue-violet excitation) and U-MWU (Ex. 330–385 nm band-pass, Em. ≥420 nm longpass, Dichroic Mirror 400 nm; UV-A excitation). Fluorescence micrographs of the larvae were taken using an ultrasensitive charge-coupled device RGB-color camera (DP-73, Olympus, Tokyo, Japan). The fluorescence micrographs of one to three-day-old larvae were taken with an exposure time of 500 ms because the fluorescence of these larvae was extremely low (one-day-old *A. tenuis* was still in the embryo stage.) On the contrary, exposure time was fixed at 50 ms without any digital black balances to visually confirm changes in larval fluorescence on four-day-old to 21-day-old larvae. Larval fluorescence spectra were measured using a Photonic Multi-channel Analyzer (PMA-C7473, Hamamatsu Photonics K.K., Shizuoka, Japan) attached to the microscope's camera port. The measurement area was set around the mouth region of the larvae (approximately the center of Figs 3 and 5), and the exposure time was set at 20,000 ms. Since the fluorescence intensity could change if the larval thickness varied, larval thickness was kept constant by placing two hairs between the glass slide and the coverslip. Three larvae each were sacrificed and measured on each observation day. The background away from the larvae was also measured. To standardize the fluorescence values, we also measured fluorescence reference slides (Fluor-Ref, Microscopy Education, www.microscopyeducation.com, "GREEN" for observations with blue-violet excitation and "UV-BLUE" for UV-A excitation). Fluorescence reference slides emit strong fluorescence and cause saturation of the PMA with the settings used. Thus, a no. 25 neutral-density filter was placed on the slides to reduce fluorescence. We calculated correction factors using the fluorescence values of the fluorescence reference slides to standardize the larval fluorescence. The peak fluorescence of GREEN under blue-violet was between 502 and 512 nm, and that of

UV-BLUE under UV-A was between 429 and 439 nm. Correction factors were calculated as follows:

Factor (blue-violet) = (average values of larval fluorescence at 502–512 nm) / (average values of fluorescence reference slide fluorescence at 502–512 nm).

Factor (UV-A) = (average values of larval fluorescence at 429–439 nm) / (average values of fluorescence reference slide fluorescence at 429–439 nm).

Fluorescence values were then multiplied by these correction factors. All observations were conducted without fixations, i.e., fluorescent micrographs and fluorescence spectra were obtained from live larvae.

## Acquisition of Symbiodiniaceae cells by *A. tenuis* larvae under red LED light

An infection experiment under red LED light was conducted to determine whether only visible larval fluorescence affects symbiodiniacean cell acquisition. Visible spectrum fluorescence, including green fluorescence, cannot be excited under red LED light. For this observation, we prepared another batch of *A. tenuis* larvae following the method published by Suzuki [25]. Ten parental colonies were collected from Sekisei lagoon, and sampling of parental corals was permitted by the Okinawa Prefectural Government for research use (No. 24–54). Five individual six-day-old aposymbiotic larvae were put into each of six 100-mL glass cups (columnar form; inner diameter = 47 mm, height = 54 mm) with 50 mL of 0.4-μm-filtered seawater. Fifty cells of cultured strains AJIS2-C2 (*S. microadriaticum*) or CCMP 2556 (*D. trenchii*) were added to the cups in triplicate. This cell density (1000 cells/L) is similar to that in the natural reef environment [10], and even at such a low cell density, *A. tenuis* larvae can acquire these native endosymbionts [14]. All experimental cups were then placed under a red LED light source (CWL; 660 nm, FWHM; 20 nm, EYELA TOKYO RIKAKIKAI CO., LTD, Tokyo, Japan) from 07:00 to 20:00. Subsequently, all larvae, without fixation, were observed under an epifluorescence microscope (BX50; filter cube U-MNV; Ex. 400–410 nm, Em. ≥455 nm longpass) to count the acquired symbiodiniacean cells within the larval body, as previously described by Yamashita et al. [14].

## Statistical analysis

All the statistical tests were performed using R version 3.6.3 [26]. To identify the relationships between light wavelength and positive/negative phototaxis of each cultured strain, we used the generalized additive model (GAM) in the mgcv package, version 1.8.33 [27]. In this modeling, the increase/decrease in the cell numbers near grids attached to each wavelength band-pass filter was assumed to follow a Gaussian distribution. Filters of 330–385 nm and 400–410 nm were considered to be 358 nm and 405 nm, respectively. A smoothing function was used for wavelength with dimension on the basis (k) = 7.

For the green fluorescence of *A. tenuis* larvae under blue-violet excitation light, we performed a likelihood ratio test (LRT) based on a generalized linear model with mixed effects (GLMM) to examine whether the peak intensity of the green fluorescence significantly changed with larval age, using the glmmTMB package, version 1.0.1 [28]. In the present study, the peak of larval green fluorescence was detected between 521 nm and 524 nm; thus, for statistical modeling, fluorescence intensity at 522.4 nm (around the middle) was used. The intensity was assumed to follow a gamma distribution, and the link function was log; the explanatory variable was larval age; the fluorescence intensity of the fluorescence reference slides (GREEN) at 522.4 nm was used as an offset. The random effect was assumed to vary among individual larvae; since, the fluorescence intensity of individual larvae may be affected by the individual-specific characteristics, such as development speed and/or parental combinations. We

prepared a model excluding the explanatory variable (larval age) for the null model. LRT was conducted between the model and null model using the anova function in R. We also conducted LRT based on GLMM to examine whether the broad blue fluorescence's peak intensity under UV-A excitation changed with larval age. In this model, larval fluorescence intensity at 463.1 nm (around the middle) was used, and the intensity of fluorescence reference slides (UV-BLUE) corresponding to this wavelength was used as an offset. Other components were the same as those used in the model for green fluorescence. The intensity values were large numbers; thus, logarithmic values were used.

We conducted LRT based on GLMM for analysis of the infection test. The percentage of Symbiodiniaceae cells acquired larvae (= infection rates) was assumed to follow a binomial distribution, and the link function was logit. The explanatory variable was the Symbiodiniaceae species (*S. microadriaticum* or *D. trenchii*). Since the infection rates may be affected by the environments in the individual experimental glass cup, the random effect was assumed to vary among the experimental glass cups. For the acquired symbiodiniacean cell densities within the exposed individual larva, the statistical test assumed a Poisson distribution of cell numbers and adopted a log link function. Other components were the same as the model for infection rates. LRT was conducted between these models and the null model, excluding the explanatory variable (supplied Symbiodiniaceae species) using the anova function in R. In the present study, $p$-values $< 0.05$ were considered to be statistically significant.

## Results

### Phototaxis of cultured Symbiodiniaceae

The phototaxis patterns of the experimental cultured strains are shown in Fig 2, and the raw count data are shown in S1 Dataset. From GAM analysis, a significant relationship between

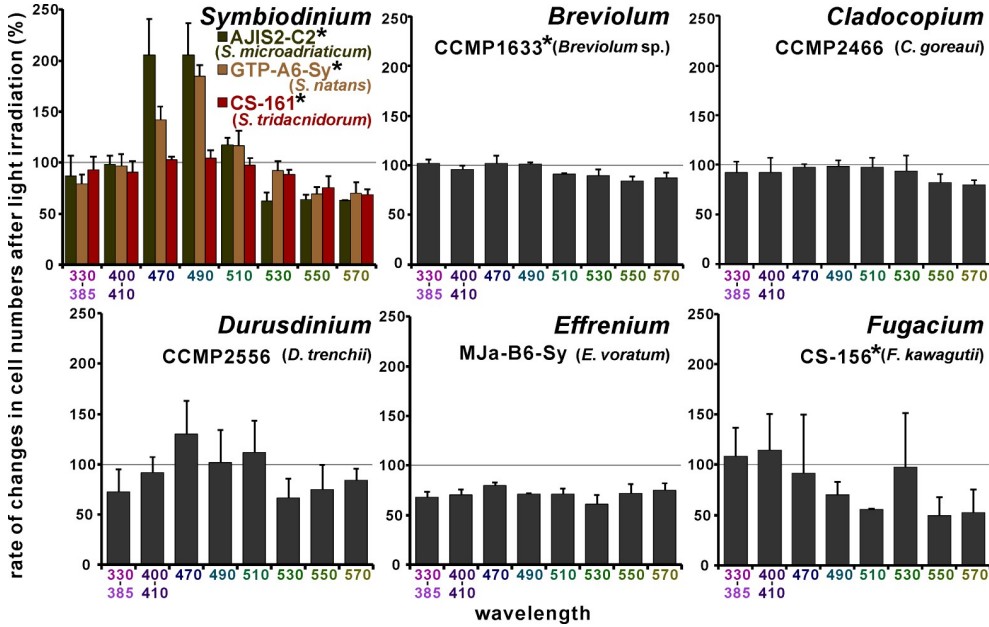

**Fig 2. The phototaxis patterns of cultured Symbiodiniaceae strains.** The 100% line indicates constant cell densities before and after irradiation. The mean and standard deviation of triplicate measurements (error bar) are shown. Asterisk on the shoulder of the strain name indicates significant relationship between wavelength and positive/negative phototaxis from GAM analysis.

wavelength and positive/negative phototaxis was found in the cultures of *S. microadriaticum* (estimated degree of freedom (edf) = 5.448, F = 26.1, adjusted $R^2$ (adj. $R^2$) = 0.868, $p < 2.0 \times 10^{-16}$), *S. natans* (edf = 5.663, F = 18.16, adj. $R^2$ = 0.821, $p = 1.93 \times 10^{-6}$), *S. tridacnidorum* (edf = 2.807, F = 8.946, adj. $R^2$ = 0.567, $p = 0.00047$), *Breviolum* sp. (edf = 3.627, F = 6.115, adj. $R^2$ = 0.516, $p = 0.00213$), and *F. kawagutii* (edf = 1, F = 6.846, adj. $R^2$ = 0.203, $p = 0.0158$). The estimated relationship between wavelength and the increase/decrease in the cell numbers of these cultured strains in the fitted GAM are also shown in S1 Fig. In the fitted GAM, the predicted peaks of the positive phototaxis action spectrum for *S. microadriaticum*, *S. natans*, and *S. tridacnidorum* were 477 nm, 488 nm, and 468 nm respectively. Negative phototaxis at the longer wavelength region was observed in *F. kawagutii*; thus, the predicted peak of the positive phototaxis action spectrum for this strain was 358 nm. In *Breviolum* sp., weak double peaks for the positive phototaxis action spectrum (358 nm and 461 nm) were predicted. The relationship between the wavelength and phototaxis was statistically significant in the cultured strains of *S. tridacnidorum*, *Breviolum* sp., and *F. kawagutii*. However, we did not observe strong positive phototaxis for a specific wavelength in these species (Fig 2). Conversely, in another native endosymbiont, *D. trenchii*, although the cells appeared to have a positive phototaxis to 470 nm light (Fig 2), a statistically supported relationship could not be described (edf = 2.074, F = 1.261, adj. $R^2$ = 0.115, $p = 0.266$). No significant relationship was found between the wavelength and phototaxis in both *C. goreaui* and *E. voratum* cultured strains (edf = 2.006 and 3.048, F = 2.485 and 0.389, adj. $R^2$ = 0.175 and 0.0251, $p = 0.125$ and 0.681, respectively).

## Fluorescence of *Acropora tenuis* larvae

In the *A. tenuis* genome, at least three cyan-green and three red fluorescent protein-like protein-encoding genes were found (S1 Appendix). These six-fluorescent protein-like proteins are also detected in aposymbiotic *A. tenuis* larvae as peptides during proteomic analysis (S1 Appendix).

Fluorescence micrographs of *A. tenuis* larvae under blue-violet excitation are shown in Fig 3. Greyscale images of the separated color channels (red, green, and blue) are also shown in S2 Appendix. Although the fluorescence intensity was low, orange fluorescence was observed after two days. The fluorescence gradually increased in intensity, and fluorescence micrographs could be taken with a 50-ms exposure time after the larvae were four days old. Green fluorescence could be visibly identified in the larvae after they were more than four days old. Larval fluorescence spectra under blue-violet excitation is shown in Fig 4. Although the PMA measured wavelength from 196 to 958 nm, data from 400 to 800 nm are plotted in Fig 4. Fluorescence at wavelengths shorter than 455 nm is almost undetectable due to the dichroic mirror in the U-MWBV2 filter cube. The raw data with and without correction by Fluor-Ref are shown in S2 Dataset. Although transmittance between 455 nm and 478 nm was less than 50% owing to characteristics of the emission filter (https://www.olympus-lifescience.com/en/optics/mirror-units/), a shoulder was found between 467 nm and 507 nm (Fig 4). The observed orange fluorescence in Fig 3 was broad with a peak at around 620 nm. Larval green fluorescence showed a peak wavelength of about 522 nm. The intensity of this green fluorescence significantly differed by larval age (Δdeviance = 40.58, Δdf = 1, $p = 1.885 \times 10^{-10}$). This can also be visually confirmed in Fig 3. The peak of the broad orange fluorescence was shifted to a shorter wavelength based on larval age (622 nm at 4 days old to 584 nm at 21 days old). Larval orange color also appears to be gradually fading (Fig 3).

The fluorescence micrographs of larvae taken under UV-A excitation are shown in Fig 5. Original figures, except those for one to three-day-old larvae, were not bright enough (S2 Fig); thus, the brightness of micrographs for larval aged four days old and over was increased in Fig 5. The greyscale images of the separated color channels (red, green, and blue) are also shown

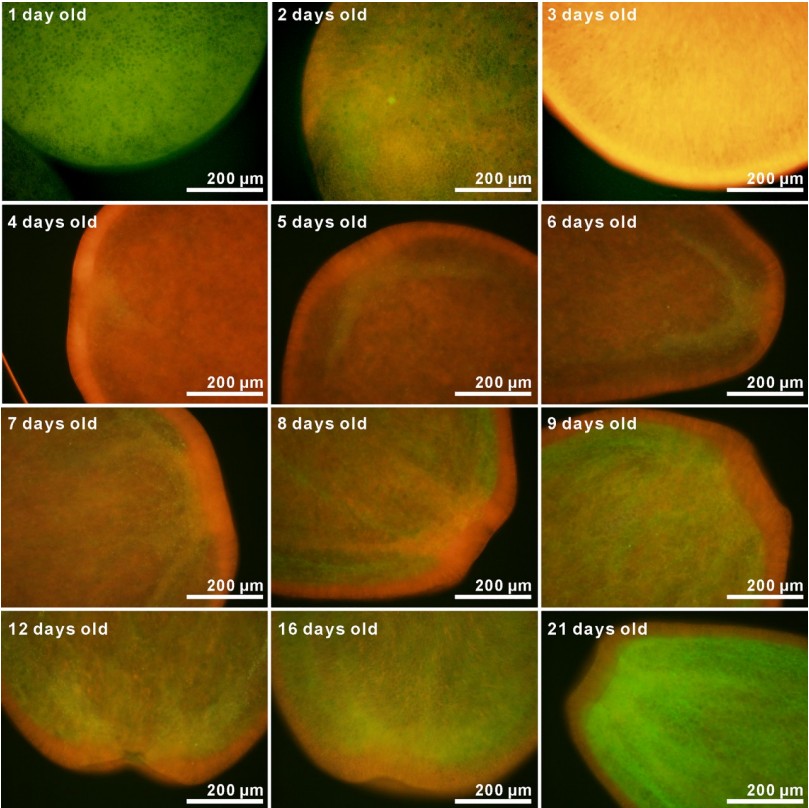

**Fig 3. Fluorescent micrographs of *A. tenuis* larvae taken under blue-violet excitation (Ex. 400–440 nm, Em. ≥475 nm).** Micrographs of one to three-day-old larvae were taken with an exposure time of 500 ms, whereas those for four-day-old or older larvae were taken with an exposure time of 50 ms and without any digital black balances. Although micrographs for one to three-day-old larvae had a slightly different color than those for four-day-old or older larvae, changes in larval fluorescence at four- to 21-days-old could be confirmed visually.

in S2 Appendix. Although the color was slightly different due to the digital black balance, blue fluorescence was observed even in the one-day-old coral. Orange fluorescence was observed in larvae older than three days. Larval fluorescence spectra under UV-A excitation is shown in Fig 6 (data from 400–800 nm are plotted). Fluorescence at wavelengths shorter than 400 nm is almost undetectable due to the dichroic mirror in the U-MWU filter cube. Raw data with and without corrections are shown in the S3 Dataset. Owing to the characteristics of the emission filter, transmittance between 400 nm and 420 nm was less than 50% (https://www.olympus-lifescience.com/en/optics/mirror-units/). Fluorescence intensities under UV-A excitation were overall lower than those under blue-violet excitation. However, a broad blue fluorescence (peak at around 450–477 nm) was recognized. The intensity of this broad blue fluorescence did not change significantly with larval age ($\Delta$deviance = 2.957, $\Delta$df = 1, $p$ = 0.08551). Orange fluorescence was also emitted; however, the 522 nm green fluorescence peak observed under blue-violet excitation was not detected.

## Acquisition of Symbiodiniaceae cells by *A. tenuis* larvae under red LED light

We observed 15 *A. tenuis* larvae supplied with either *S. microadriaticum* or *D. trenchii* (5 larvae / cup × three cups each). The percentages of *S. microadriaticum* infected larvae in each cups under red LED light were 0%, 0%, and 20%, but the percentages of *D. trenchii* infected larvae under red

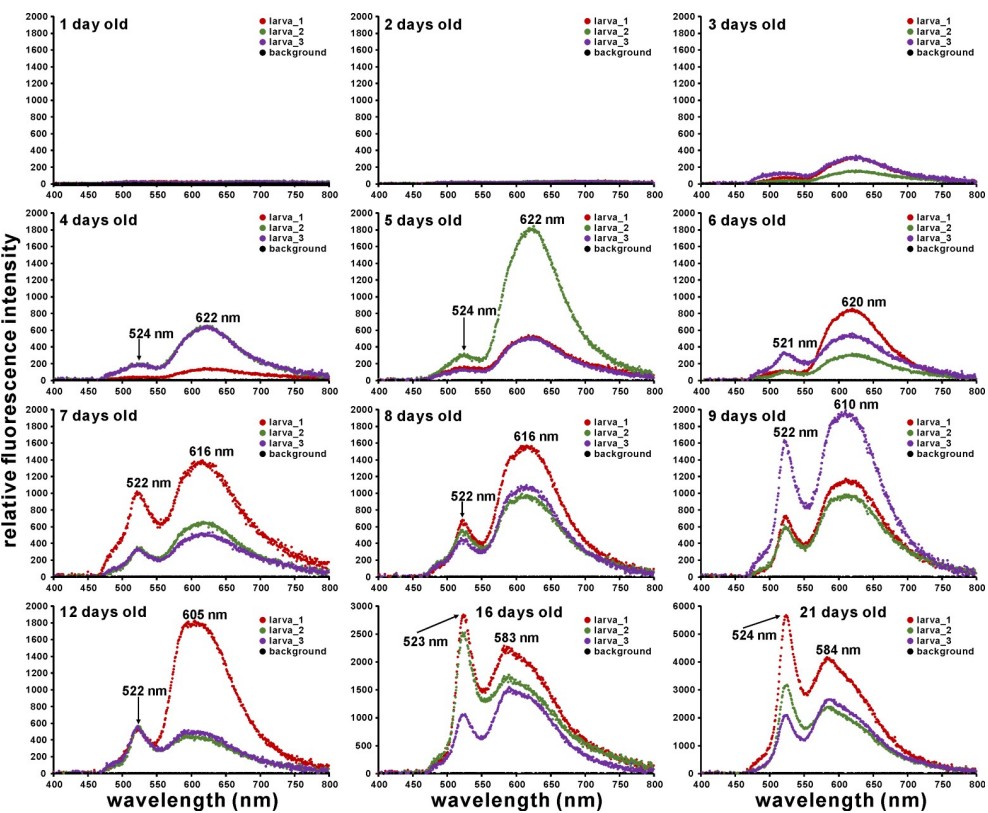

**Fig 4. Fluorescence spectrum of *A. tenuis* larvae under blue-violet excitation (Ex. 400–440 nm, Em. ≥475 nm).**
Peak wavelength at which the maximum intensity was recorded is shown in each graph. Fluorescence intensity in one-day-old and two-day-old larvae was almost at the same level as in the background. The y-axis values for the 16- and 21-day-old larvae are different from the others.

LED light were 100%, 40%, and 40% (Fig 7). The percentage of Symbiodiniaceae cell infected larvae significantly differed depending on the supplied species (Δdeviance = 5.6493, Δdf = 1, $p$ = 0.01746). The number of *S. microadriaticum* cells acquired within each larva under red LED light was one cell/larva (only one infected larva with one *S. microadriaticum* cell). Acquired *D. trenchii* cell number in larva was 1.2 ± 0.22 cells/larva. The numbers of symbiodiniacean cells acquired by the larvae significantly differed depending on the supplied species (Δdeviance = 5.8, Δdf = 1, $p$ = 0.01603).

## Discussion

The question of how new generations of corals and appropriate Symbiodiniaceae species connect in the early stages of symbiosis has long been debated. The beacon hypothesis [18, 22] that aposymbiotic larvae attract symbiodiniacean cells using GFP is interesting. However, information is lacking on phototaxis action spectrum of native Symbiodiniaceae symbionts and fluorescence spectrum of the aposymbiotic larvae at the time of endosymbiont acquisition.

### Phototaxis patterns of Symbiodiniaceae cells

Phototaxis patterns differ among Symbiodiniaceae species. Cultured native endosymbiont species *S. microadriaticum* displayed positive phototaxis to 470 nm and 490 nm of blue light

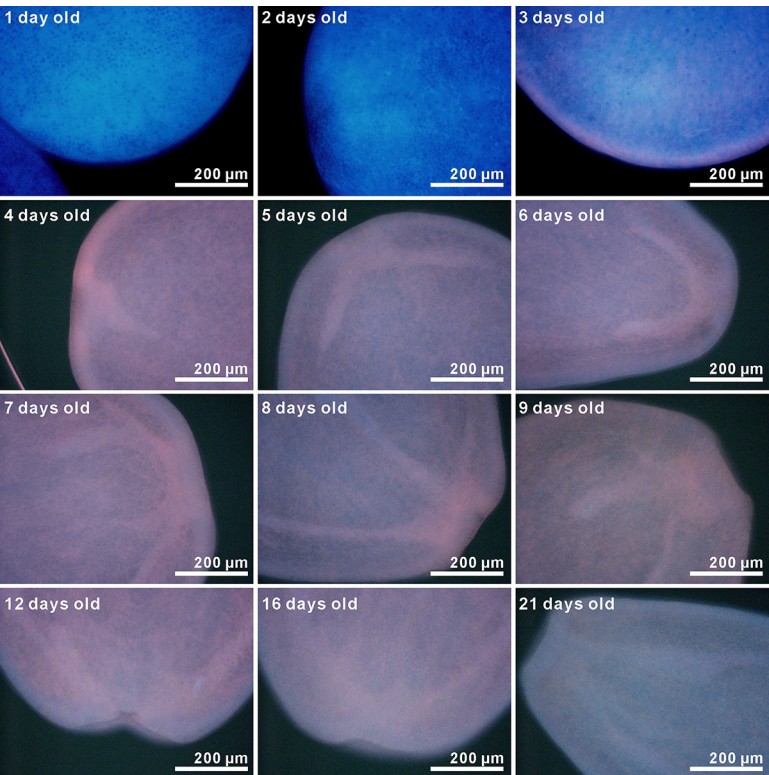

**Fig 5. Fluorescent micrographs of *A. tenuis* larvae taken under UV-A excitation (Ex. 330–385 nm, Em. ≥420 nm).**
The micrographs of one to three-day-old larvae were taken with an exposure time of 500 ms, whereas those for four-day-old or older larvae were taken with an exposure time of 50 ms and without any digital black balances. Although, the micrographs of larvae aged one to three days and those older than four days are slightly different in color, changes in larval fluorescence between four-day-old larvae and 21-day-old larvae can be confirmed visually. The brightness of micrographs for larvae aged four days old and over are increased. The original figures are shown in S2 Fig.

(Fig 2). A similar trend was also found in a cultured strain of the free-living species *S. natans*, although the predicted peak of positive phototaxis action spectrum was different (477 nm for *S. microadriaticum* and 488 nm for *S. natans* in fitted models, S1 Fig). Although a shorter wavelength was not observed in the present study, this tendency was almost congruent with some dinoflagellates that showed highest phototaxis peak at blue light region (e.g., [29]). The motile stage of Symbiodiniaceae has an eyespot composed of crystalline layers forming a quarter-wavelength film-like structure, and calculations based on the thickness of each layer indicate that it is ideal for reflection of light with wavelengths of 320–520 nm [23]. Thus, phototaxis patterns observed in these species are reasonable considering eyespot structure. However, a clear positive phototaxis pattern was not found in another native endosymbiont *D. trenchii* strain. Further, in the green light region (510–570 nm), strong positive phototaxis was not found in any tested species (Fig 2).

## Fluorescence spectrum of *A. tenuis* larvae under blue-violet excitation

Green fluorescence with a peak around 521–524 nm was detected by PMA from > three-day-old larvae under blue-violet (400–440 nm) excitation light, and this fluorescence significantly increased each day (Fig 4). In laboratory experiments, *A. tenuis* larvae begin to acquire symbiodiniacean cells at around three to four days of age [14, 30]. The coincidence of the timing of endosymbiont acquisition and GFP concentration around the larval mouth was previously

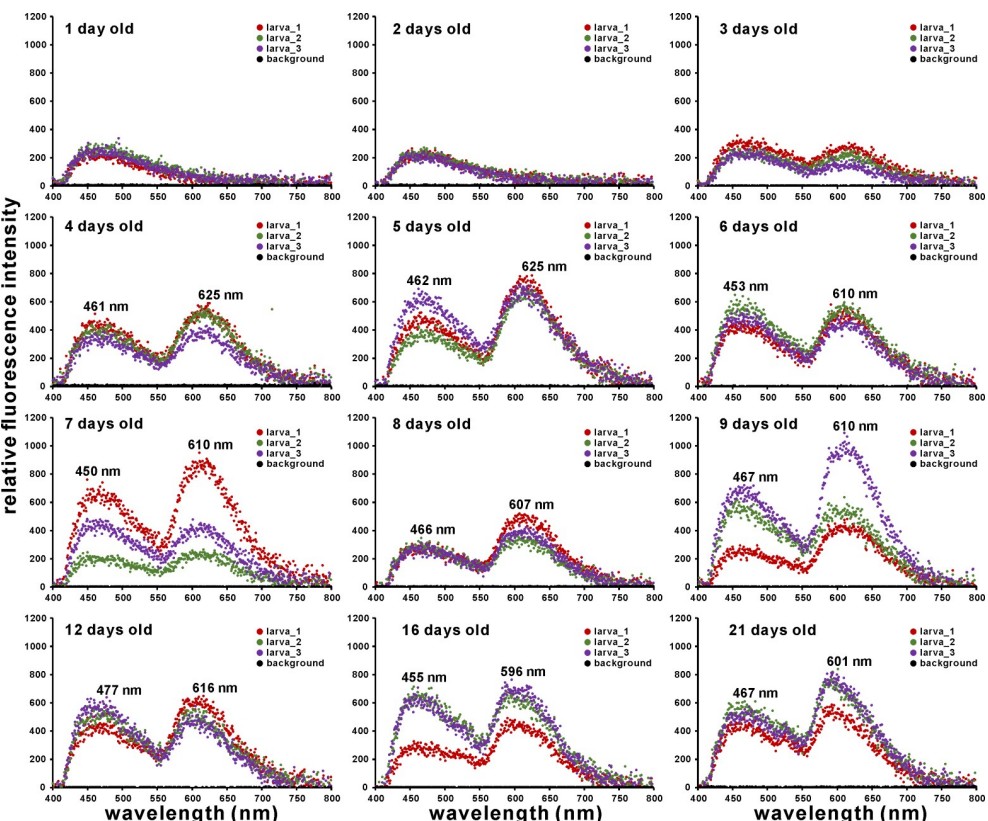

**Fig 6. Fluorescence spectrum of *A. tenuis* larvae under UV-A excitation (Ex. 330–385 nm, Em. ≥420 nm).** Peak wavelength at which the maximum intensity was recorded is shown in each graph.

reported in *Fungia scutaria* [22]. Aihara et al. [20] demonstrated that 1) green morph *E. aspera* corals emit green fluorescence with a peak at 505 nm under blue excitation, and 2) cultured symbiodiniacean strain OTcH-1 shows the highest phototaxis activity toward this region of green light (peak at 510 nm). These observations make the beacon hypothesis more believable. However, strain OTcH-1 was originally isolated from giant clam *Tridacna crocea* [31], and belongs to "type A6" based on ITS2 sequence (GenBank accession number AB097464) and Symbiodiniaceae ITS2 sequence database "GeoSymbio" [32] (https://sites.google.com/site/geosymbio/). Type A6 and A3 Symbiodiniaceae are now together classified into single species, *S. tridacnidorum* [33], but type A6 endosymbionts are usually not found in adult corals [33, 34] or in juvenile corals [14, 35]. A simple comparison is difficult due to the different photo-taxis measurements; still, phototaxis pattern of our *S. tridacnidorum* cultured strain (CS-161; type A3) was not consistent with those reported by Aihara et al. [20]. At a wavelength of larval green fluorescence (peak around 522 nm), the native endosymbiont *S. microadriaticum* is pre-dicted to show slightly negative phototaxis (S1 Fig and also Fig 2, at 530 nm). If this endosym-biont focuses only on larval green fluorescence, this could have a negative effect. Conversely, the shoulder of the larval fluorescence (between 467 nm and 507 nm; around the cyan region) covered the positive phototaxis action wavelength of the native endosymbiont *S. microadriaticum*.

A broad orange fluorescence also appeared under blue-violet excitation (Figs 3 and 4). This orange fluorescence is expected to be from merged multiple fluorescence peaks since multiple genes encoding fluorescent proteins are found within *Acropora* corals [36]. In our

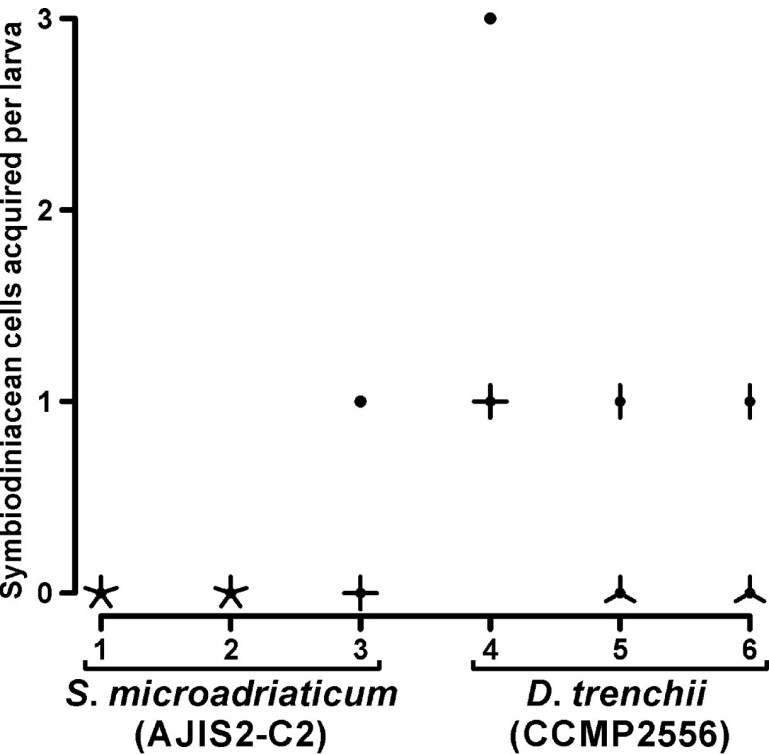

**Fig 7. The symbiodiniacean cell numbers within larvae under red LED light source.** Acquired cell numbers are shown in a sunflower plot for each experimental cup separately. Number of leaves (petals) indicating that individual larvae harbored the same numbers of symbiodiniacean cells. Five individual six-day-old larvae were observed in each experimental cup.

experimental coral *A. tenuis*, we also found multiple fluorescent protein-encoding genes (S1 Appendix). Further, some fluorescent proteins could be detected in *A. tenuis* larvae (S1 Appendix). The peak of the orange fluorescence gradually shifted to the shorter wavelength with larval age (Fig 4). The orange fluorescence of the long-wavelength peak might be gradually decreasing. Further, we occasionally found faint orange fluorescence larvae. These larvae were visibly greenish under a fluorescent microscope (S3 Appendix). Different larval families exhibit extensive variation in gene expression [37], and larval color variation is explained partially by parental effects [38]. In the present study, we prepared larvae from nine parental *A. tenuis* colonies. Thus, differences in the intensity of orange fluorescence may be due to parental combinations. However, the peak wavelength of green fluorescence was well matched in all observed larvae (S3 Appendix).

## Fluorescence spectrum of *A. tenuis* larvae under UV-A excitation

Under UV-A (330–385 nm) excitation light, a broad blue fluorescence peak of about 450–477 nm was detected in *A. tenuis* larvae (Figs 5 and 6). Since *S. microadriaticum* showed positive phototaxis for 470 nm and 490 nm light, it is plausible to assume that larval blue fluorescence could attract this native endosymbiont. In natural coral reef environments, high water column transparency allows UV radiation to penetrate to depths of 20 m or more (e.g., [39]). Longer wavelength UV radiation (UV-A; 315–400 nm) can reach farther than the shorter wavelength of UV radiation (UV-B; 280–315) (e.g., [40]). This phenomenon is also observed in coral reef areas (e.g., [41, 42]). Negative effects of UV radiation on organisms are often recognized;

however, UV-A plays a minor or insignificant role in egg/larvae survivorship in broadcast-spawning corals [43]. Thus, larvae might utilize UV-A radiation. Natural *Acropora* larvae commonly settle in shaded spots on the substratum (e.g., gaps, crevices, grooves), probably because they are usually under pressure from chronic grazing of some fish and benthic invertebrates (e.g., [44, 45]). The light environment in these shaded spots is thought to consist of mainly scattered and reflected light rather than direct sunlight. Larvae/juveniles may be able to attract *S. microadriaticum* if a dull broad blue fluorescence is emitted in these dim locations.

## Acquisition of native Symbiodiniaceae cells under red LED light

We used two cultured strains of native endosymbionts, AJIS2-C2 (*S. microadriaticum*) and CCMP2556 (*D. trenchii*). In previous observations, these cultured strains were attracted to and acquired by *A. tenuis* larvae under normal white light [14]. In the present study, infection tests were conducted under red LED light. Larval visible spectrum fluorescence, including green fluorescence, cannot be excited with this light. Almost no larvae (14 of 15 observed larvae) acquired *S. microadriaticum* cells; however, 9 of 15 observed larvae acquired *D. trenchii* cells (Fig 7). In the present study, clear positive phototaxis to specific wavelength light was not found for the *D. trenchii* cultured strain. However, the members of the genus *Durusdinium*, including *D. trenchii*, are often found within *Acropora* larvae and juveniles [10, 13, 14, 46, 47]. These results indicate that not only larval fluorescence, but also other factors are involved in the attraction of *D. trenchii* cells. Some Symbiodiniaceae cells can be attracted by aposymbiotic host animals [15] or their extracts [19], and "attractants" are a variety of nitrogen-containing compounds [16]. Recently, Takeuchi et al. [17] demonstrated that the N-acetyl-D-glucosamine (GlcNAc)-binding fraction of *A. tenuis* extracts is the most plausible candidate for a chemoattractant. In fact, *D. trenchii* cells can be attracted to this lectin [48].

   *S. microadriaticum* cells were not acquired by *A. tenuis* larvae under red LED light. *A. tenuis* larvae can acquire this native endosymbiont under normal white light [14] or blue LED light (CWL; 470 nm, FWHM; 30 nm; S4 Appendix). In *Acropora millepora* juveniles, Quigley et al. [35] found that genus *Symbiodinium*, including *S. microadriaticum* endosymbionts, occurred at a significantly higher abundance in redder juveniles compared to greener juveniles. The visibly greener *A. tenuis* larvae under fluorescent microscopy (Ex. 400–440 nm, Em. ≥475 nm; blue-violet excitation) have less orange fluorescence (S3 Appendix). Given these results, *S. microadriaticum* cells may be attracted to the orange fluorescence of larvae or juveniles. Nonetheless, the current phototaxis analysis was conducted using limited wavelengths because we focused on whether native endosymbionts possess positive phototaxis to green light. Thus, further analysis is required to conclude on the above-proposed mechanisms.

   In *Acropora* corals, after mass-spawning and subsequent fertilizations, the larvae initially swim in the water column and then settle in shaded spots on the substratum. Thus, various factors, including spacious distances and the effect of water flow, should be taken into account between or around aposymbiotic corals and their native symbiodiniacean symbionts in the environments. A chemotaxis experiment using polyps of the soft coral *H. fuscescens* and their endosymbionts demonstrated that the endosymbiont cells swam toward the polyp against flow velocities of up to 0.5 mm s$^{-1}$; however, the cells must be within a distance in the order of a centimeter to decimeter from the polyp to sense the chemoattractant due to the dispersion of the substance [49]. In fluorescent attraction, Aihara et al. [20] demonstrated that 8-mm$^2$ *E. aspera* coral fragments can attract symbiodiniacean cells within a 95-mm diameter plastic dish. However, it is still unclear how far *A. tenuis* larvae/juveniles can attract native endosymbiont species using chemical and/or fluorescent attractants in the natural environments.

## Conclusions

In the present study, the fluorescence spectrum of *A. tenuis* larvae and phototaxis of their native endosymbionts were measured simultaneously. Phototaxis patterns differ among Symbiodiniaceae species. The native endosymbiont, *S. microadriaticum* can be attracted strongly to 470–490 nm. However, another native endosymbiont, *D. trenchii*, did not show clear positive phototaxis in the present study. Under blue-violet excitation light, the peak of green fluorescence in *A. tenuis* larvae was about 522 nm. Although the shoulder of this fluorescence spectrum (cyan region) covered the positive phototaxis action wavelength of *S. microadriaticum*, the maximum green fluorescence peak and maximum phototaxis action wavelength did not coincide in our observations. At the peak wavelength of larval green fluorescence, *S. microadriaticum* showed slightly negative phototaxis. Conversely, under UV-A excitation light, a broad blue fluorescence was observed in *A. tenuis* larvae. This blue fluorescence coincided well with the positive phototaxis action wavelength of *S. microadriaticum*. We also performed infection tests using *A. tenuis* larvae and cultured strains of *S. microadriaticum* and *D. trenchii*, under red LED light sources. The results demonstrated that there might be attraction mechanisms other than those induced by visible fluorescence. This scenario is plausible, because fluorescence is not an exclusive characteristic of larvae/juvenile. A variety fluorescence spectra, not limited to green fluorescence, have been reported in adult corals (e.g., [50–52]). Aihara et al. [20] clearly demonstrated that some Symbiodiniaceae cells are attracted to the green morph *E. aspera* and even toward green fluorescent dye. This result indicates that symbiodiniacean cells could be attracted indiscriminately if phototaxis action wavelength and fluorescence spectra are well matched. We found a blue peak at 480~486 nm and a green peak at 512 nm fluorescence in adult *A. tenuis* colonies (S5 Appendix). The blue fluorescence spectra matched well with the maximum positive phototaxis action wavelength of one of the native endosymbionts of larvae/juveniles. Thus, if aposymbiotic larvae/juveniles attract suitable Symbiodiniaceae cells using only fluorescence, it may be difficult to encounter endosymbiont candidates, because larval/juvenile fluorescence would be overwhelmed by fluorescence of adult corals. However, aposymbiotic *A. tenuis* larvae certainly attracted native endosymbionts in the proximity of laboratory experimental levels [14]. Although the effective distances of these attractions are still unclear, if the native endosymbiont candidates are within or near the small shaded locations on substrates where the larvae settle, these attractions might be possible in the environment. Other factors could also be considered. For example, fluorescence variation could be associated with preferential uptake of Symbiodiniaceae types [35]. In the case of soft coral, motile phase cells of endosymbiont are attracted to solutions containing exudates of aposymbiotic juvenile but not adult polyps [19]. These results could potentially provide important information to clarify the mechanisms underlying the specific attraction of certain Symbiodiniaceae types or species toward larvae/juvenile corals. New generations of coral are considered to have multiple Symbiodiniaceae cell attraction mechanisms and the ability to use combinations of these mechanisms for attracting native Symbiodiniaceae cells. To elucidate the initial attraction mechanisms, comprehensive research that includes not only green fluorescence but also other fluorescence and chemical attractants is needed.

## Supporting information

**S1 Fig. Smoothed relationships between wavelength and positive/negative phototaxis of five cultured strains (AJIS2-C2, GTP-A6-Sy, CS-161, CCMP1633, and CS-156).** A significant relationship was not found in the remaining three cultured strains (CCMP2466, CCMP2556, and MJa-B6-Sy). The y-axis represents the smoothed function with estimated degrees of freedom in parenthesis. Whiskers on x-axis indicated observed wavelengths. Filters

330–385 nm and 400–410 nm are considered as 358 nm and 405 nm, respectively. The solid line indicates estimated smooth functions and dashed lines indicates 95% confidence intervals. (TIF)

**S2 Fig. Original fluorescent micrographs of *A. tenuis* larvae taken under UV-A excitation (330–385 nm).** Micrographs with increased brightness are shown in Fig 5 in the main text. (TIF)

**S1 Appendix. Fluorescent protein-like proteins in *A. tenuis*.** (DOCX)

**S2 Appendix. Greyscale images of separated color channels (red, green, and blue) of Figs 3 and 5.** (DOCX)

**S3 Appendix. Fluorescent micrographs and fluorescence spectra of faint orange fluorescence *A. tenuis* larvae (greenish larvae).** (DOCX)

**S4 Appendix. Acquisition of native Symbiodiniaceae cells by *A. tenuis* larvae under blue LED light.** (DOCX)

**S5 Appendix. Fluorescence spectra of three adult *A. tenuis* colonies under blue-violet (400–440 nm) and UV-A (330–385 nm) excitation.** (DOCX)

**S1 Dataset. Raw Symbiodiniaceae cell count data.** This data was used in Figs 2 and S1. (XLSX)

**S2 Dataset. Raw fluorescence spectral data from *A. tenuis* larvae and adult colonies taken under blue-violet (400–440 nm) excitation light.** These data were used for Fig 4, and S3 and S5 Appendices. (XLSX)

**S3 Dataset. Raw fluorescence spectral data from *A. tenuis* larvae and adult colonies taken under UV-A (330–385 nm) excitation light.** These data were used for Fig 6 and S3 and S5 Appendices. (XLSX)

## Acknowledgments

The authors thank the staff at the Yeyama Field station, Fisheries Technology Institute, Japan Fisheries Research and Education Agency for their support during this investigation. The authors would like to thank Enago (www.enago.jp) for the English language review.

## Author Contributions

**Conceptualization:** Hiroshi Yamashita, Kazuhiko Koike.

**Data curation:** Hiroshi Yamashita.

**Formal analysis:** Hiroshi Yamashita.

**Funding acquisition:** Hiroshi Yamashita, Kazuhiko Koike, Chuya Shinzato.

**Investigation:** Hiroshi Yamashita, Chuya Shinzato, Mitsuru Jimbo.

**Methodology:** Hiroshi Yamashita.

**Project administration:** Hiroshi Yamashita.

**Resources:** Hiroshi Yamashita, Kazuhiko Koike, Chuya Shinzato, Mitsuru Jimbo, Go Suzuki.

**Supervision:** Hiroshi Yamashita.

**Validation:** Hiroshi Yamashita.

**Visualization:** Hiroshi Yamashita.

**Writing – original draft:** Hiroshi Yamashita.

**Writing – review & editing:** Hiroshi Yamashita, Kazuhiko Koike, Chuya Shinzato, Mitsuru Jimbo, Go Suzuki.

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
