## [Decision Letter · Decision Letter 0]

29 Sep 2020

PONE-D-20-25327

Possible attraction of native symbiodiniacean species to Acropora tenuis larvae caused by non-green fluorescence factors

PLOS ONE

Dear Dr. Yamashita,

Thank you for submitting your manuscript to PLOS ONE. After careful consideration, we feel that it has merit but does not fully meet PLOS ONE’s publication criteria as it currently stands. Therefore, we invite you to submit a revised version of the manuscript that addresses the points raised during the review process.

We look forward to receiving your revised manuscript.

Kind regards,

Anderson B. Mayfield, Ph.D.

Academic Editor

PLOS ONE

Journal Requirements:

2. In your Methods section, please provide additional location information of the sampling sites, including geographic coordinates for the data set if available.

Reviewers' comments:

Reviewer's Responses to Questions

**Comments to the Author**

1. Is the manuscript technically sound, and do the data support the conclusions?

Reviewer #1: Partly

Reviewer #2: Partly

2. Has the statistical analysis been performed appropriately and rigorously? 

Reviewer #1: No

Reviewer #2: No

3. Have the authors made all data underlying the findings in their manuscript fully available?

Reviewer #1: Yes

Reviewer #2: Yes

4. Is the manuscript presented in an intelligible fashion and written in standard English?

Reviewer #1: Yes

Reviewer #2: Yes

5. Review Comments to the Author

Reviewer #1: This manuscript entitled “Possible attraction of native symbiodiniacean species to Acropora tenuis larvae caused by non-green fluorescence factors” is a very interesting study and provides results I think are worthwhile to the greater coral community as a whole. I overall enjoyed the scope and experiments the paper presented. There is substantial lack of data on the potential for symbiodiniacean phototaxis so this study begins to fill in that gap. My major concerns for this paper are that the samples sizes are insufficient for the majority of the experiments performed. Especially since it is known there is substantial genotypic and phenotypic variability with regards to coral larval fluorescence (Meyer et al 2011, Kenkel et al 2011, etc.). Therefore, the low samples sizes here (3 larvae for photographs, 3 juveniles, 3 adults, no replicates for the infection experiment) does not lend me to trust these results are representative of phenotypes in this system. Further, there was not sufficient replication for the symbiodiniacean phototaxis experiments (were different cohorts of symbiodinaceans examined? on different days?) as well and some clarity on that is warranted. The biggest concern is the lack of any statistical test on the phototaxis experiments (Figure 2) from what I can tell. Further, the authors deduce positive or negative phototaxis of certain types (Durisdinium for example) where I worry a statistical test would invalidate that claim considering the breadth of the error bars. While the small samples sizes are unfeasible to be changed at this conjecture, I think at least discussion or consideration of this is warranted.

Line 115-116: It would be good to review the recommendations for this journal, but typically the permit information would go at the end. It is a distraction for the reader where it is now.

Line 159: Were these three repeated measurements with different cohorts of Symbiodinaceae or the same exact sample of Symbiodinaceae? I worry there is not enough replication for this study.

Figure 1: I don’t understand how samples could have been taken from each grid if the culture flask was standing upright, as it appears in Figure 1. If you then put the flask on it’s side to take a sample of each grid wouldn’t that redistribute the symbionts within the flask?

Figure 2: The authors report positive phototaxis for more strains than what seems apparent from Figure 2. Where are the statistics reported to tests significant deviations from null expectations?

Figure 4: you cannot distinguish the colors denoting differences between larva 1, 2, and 3. The figure needs to be replotted to clarify that. In addition, the resolution appears to be too low. Further, there appears to be substantial variation in fluorescence intensity between the three larvae measured. and many times the larvae is the exact same as the background (from what I can tell). This leads me to believe there was not sufficient sample sizes to truly deduce the larval fluorescent signal from the background as well as represent the existing variation among larval cohorts.

Reviewer #2: Larvae of many reef-building corals are initially aposymbiotic and need to acquire their endosymbiotic algae horizontally, presumably from the water column, where their abundance is relatively low. It has been proposed before that green fluorescence of the larvae serves as a “beacon” to attract the algae for establishing symbiosis. The present manuscript explores the possibility that different light qualities differentially affect the behavior of the different species of Symbiodiniaceae found as endosymbionts in Acropora tenuis juveniles in the wild, as well as the possibility that different fluorescence spectra exhibited by A. tenuis larvae of different ages might enable their more efficient colonization by the algae. The authors first describe how various strains of cultured Symbiodiniaceae are attracted (or not) to light of different wavelengths. They then look at changes in the fluorescence during development of aposymbiotic larvae under violet-blue and UV-A excitation, analyze potential fluorescent-protein genes and peptides from such proteins, and examine the changes in larval, juvenile, and adult fluorescence after infection by the algal strains typically found in A. tenuis. The final experiment assesses infection of larvae by these algal strains under either blue or red LED light. Given the widespread use of cnidarian-derived fluorescent proteins in various fields of biology, it remains surprising how little is known of their function(s) in the animals from which they are derived. I think that further investigation of the hypothesis that these proteins might help coral larvae to attract suitable endosymbionts from the water column is of potentially high value. But I see several large and small flaws in the current manuscript and so recommend a major revision. Please see the attached document for the detailed comments and suggestions.

6. PLOS authors have the option to publish the peer review history of their article (what does this mean?). If published, this will include your full peer review and any attached files.

Reviewer #1: No

Reviewer #2: No

---

## [Author Response · Author response to Decision Letter 0]

24 Nov 2020

Please see our attached "Response to Reviewers" file.

---

## [Decision Letter · Decision Letter 1]

8 Jan 2021

PONE-D-20-25327R1

Can Acropora tenuis larvae attract native Symbiodiniaceae cells by green fluorescence at the initial establishment of symbiosis?

PLOS ONE

Dear Dr. Yamashita,

Thank you for submitting your manuscript to PLOS ONE. After careful consideration, we feel that it has merit but does not fully meet PLOS ONE’s publication criteria as it currently stands. Therefore, we invite you to submit a revised version of the manuscript that addresses the points raised during the review process.

We look forward to receiving your revised manuscript.

Kind regards,

Anderson B. Mayfield, Ph.D.

Academic Editor

PLOS ONE

Additional Editor Comments (if provided):

Hello,

I am pleased to say that both reviewers were impressed by your revision and ultimately would like to see this manuscript published. However, it needs professional proofreading by a native English speaker. As I am a fairly new editor, I am not sure if PLoS ONE offers such services. I myself am an English editor, but I do not want to recommend myself for fear of a conflict of interest. Perhaps you could find an English speaking colleague in your department, or maybe pay for someone to do it. I normally charge $120/manuscript, so I think it won't be too expensive. Otherwise, both reviewers have mentioned some other, smaller science-related concerns and suggestions that you could also incorporate prior to submitting the next version (do you choose to do so). I look forward to seeing it here in the next few weeks!

Anderson

Reviewers' comments:

Reviewer's Responses to Questions

**Comments to the Author**

1. If the authors have adequately addressed your comments raised in a previous round of review and you feel that this manuscript is now acceptable for publication, you may indicate that here to bypass the “Comments to the Author” section, enter your conflict of interest statement in the “Confidential to Editor” section, and submit your "Accept" recommendation.

Reviewer #1: All comments have been addressed

Reviewer #2: (No Response)

2. Is the manuscript technically sound, and do the data support the conclusions?

Reviewer #1: Partly

Reviewer #2: Yes

3. Has the statistical analysis been performed appropriately and rigorously? 

Reviewer #1: Yes

Reviewer #2: Yes

4. Have the authors made all data underlying the findings in their manuscript fully available?

Reviewer #1: Yes

Reviewer #2: Yes

5. Is the manuscript presented in an intelligible fashion and written in standard English?

Reviewer #1: No

Reviewer #2: Yes

6. Review Comments to the Author

Reviewer #1: Overall, the authors addressed and made substantial changes to the manuscript per my earlier suggestions. I appreciate that they added clarification about the sample sizes for the fluorescence measurements and included statistical tests. I still think this paper presents interesting results that should be published, although there are definitely caveats due to the difficulty of these types of experiments, which I think need more attention in the discussion. Finally, I found multiple instances of grammar and syntax mistakes in the updated manuscript, so I would recommend an editor to help fix these.

I think there should be more of a discussion of the known chemical cues involved in Symbiodiniaceae attraction to corals in the introduction. And on that note, since the results are clear that there are other factors besides fluorescence involved, integrating a discussion of these chemical signals is warranted. Finally, it is interesting that there is no phototaxis of Durisdinium, despite it’s high infectivity (not necessarily substantiated by the results here, but in other papers (Abrego, Van Open and Willis Molecular Ecology 2009, as one example). Therefore, it’s possible that different symbionts have vastly different mechanisms of detecting corals and modes of infectivity. I think the paper could be improved by discussion of these results in this broader context.

Minor comments:

Line 39: remove comma after cells.

Line 55: Confused here, since you are specifying using red LED light and you did not test for green light. If I understand you correctly, I think you should remove “including green fluorescence” from this statement.

Line 58-59: I would recommend stating the conclusions that while there is some evidence of phototaxis to light in S. microadriatricum, this phototaxis does not appear to play a role in attraction to coral larvae (at least of this species of coral). Your results also found that the high infectivity of Durisdinium was not explained by phototaxis but is likely driven by another mechanism.

Line 215: Unclear here how many colonies contributed to the genetic diversity within this new pool of larvae.

Line 350: This sentence should be adjusted for clarity. For example, “We observed 15 A. tennis individuals supplied with either S. microadriatricum or D. trenchii.”

Lines 356-358: These infection numbers are extremely low so not likely to be particularly relevant ecologically, especially since larvae can simply be eating the cells rather than taking them up into their cells. Because the numbers are so low I don’t really believe the symbionts are being acquired. I think discussion of the caveats of these results is warranted or they should be taken out.

Line 409: change to “comparison”.

Line 514-516: Does this refer to a specific study or your own results? I don’t believe the infection results presented in this study are enough to warrant this statement.

Reviewer #2: I think the manuscript from Yamashita et al. has been significantly improved both in the presentation of the data and the English; it is now much more comprehensible and easier to follow. The responses by the authors were also helpful and resolved together with the changes to the manuscript many previously criticized points and added some interesting context. The small sample sizes are still a concern, but I believe that the statistical analyses can compensate at least partially for that and the authors are now more careful in their interpretation of the data. There are still a number of major and minor points, I feel the authors need to address. Please see the attached document for the detailed comments.

7. PLOS authors have the option to publish the peer review history of their article (what does this mean?). If published, this will include your full peer review and any attached files.

Reviewer #1: No

Reviewer #2: No

---

## [Author Response · Author response to Decision Letter 1]

9 Feb 2021

Please see our attached "Response to Reviewers" file.

---

## [Decision Letter · Decision Letter 2]

4 Mar 2021

PONE-D-20-25327R2

Can Acropora tenuis larvae attract native Symbiodiniaceae cells by green fluorescence at the initial establishment of symbiosis?

PLOS ONE

Dear Dr. Yamashita,

Thank you for submitting your manuscript to PLOS ONE. After careful consideration, we feel that it has merit but does not fully meet PLOS ONE’s publication criteria as it currently stands. Therefore, we invite you to submit a revised version of the manuscript that addresses the points raised during the review process.

ACADEMIC EDITOR:

Hello, 

    I have had your article re-reviewed a third time by one of the prior reviewers, and he/she believes this article should be good to go after another round of minor revisions. As such, I believe you are in the "home stretch" with respect to having it published! Looking forward to the revisions in the coming weeks, 

Anderson

We look forward to receiving your revised manuscript.

Kind regards,

Anderson B. Mayfield, Ph.D.

Academic Editor

PLOS ONE

Journal Requirements:

Additional Editor Comments (if provided):

Hello,

I have had your article re-reviewed a third time by one of the prior reviewers, and he/she believes this article should be good to go after another round of minor revisions. As such, I believe you are in the "home stretch" with respect to having it published! Looking forward to the revisions in the coming weeks,

Anderson

Reviewers' comments:

Reviewer's Responses to Questions

**Comments to the Author**

1. If the authors have adequately addressed your comments raised in a previous round of review and you feel that this manuscript is now acceptable for publication, you may indicate that here to bypass the “Comments to the Author” section, enter your conflict of interest statement in the “Confidential to Editor” section, and submit your "Accept" recommendation.

Reviewer #2: (No Response)

2. Is the manuscript technically sound, and do the data support the conclusions?

Reviewer #2: Yes

3. Has the statistical analysis been performed appropriately and rigorously? 

Reviewer #2: Yes

4. Have the authors made all data underlying the findings in their manuscript fully available?

Reviewer #2: Yes

5. Is the manuscript presented in an intelligible fashion and written in standard English?

Reviewer #2: Yes

6. Review Comments to the Author

Reviewer #2: I think, the revised manuscript of Yamashita et al. has again much improved. The authors addressed my previous comments and concerns either in the manuscript itself or in their replies to my comments. That said, I only have one major issue which I feel needs to be addressed in the results part of the manuscript concerning the interpretation of the algal phototaxis action spectra (please see major general comment 1.). All of my other comments are mostly just corrections of some expressions and typos which the authors should be able to easily incorporate into the manuscript.

7. PLOS authors have the option to publish the peer review history of their article (what does this mean?). If published, this will include your full peer review and any attached files.

Reviewer #2: No

---

## [Author Response · Author response to Decision Letter 2]

11 Mar 2021

Please see our attached "Responses to Reviewer" file.

---

## [Editor Report · Decision Letter 3]

25 Mar 2021

PONE-D-20-25327R3

Can Acropora tenuis larvae attract native Symbiodiniaceae cells by green fluorescence at the initial establishment of symbiosis?

PLOS ONE

Dear Dr. Yamashita,

Thank you for submitting your manuscript to PLOS ONE. After careful consideration, we feel that it has merit but does not fully meet PLOS ONE’s publication criteria as it currently stands. Therefore, we invite you to submit a revised version of the manuscript that addresses the points raised during the review process.

Hello,

I am happy to see all the changes made, which have dramatically improved this manuscript. At this stage, I would only ask for clarity on a few things, as well as consider having the article proofread by a native English speaker. I have asked PLoS ONE since perhaps they provide such a service. In the mean time, here are some suggestions for improvement:

1. Please mention the nature of the error bars in the figures: standard deviation?

2. When using numbers <10, you should write them out, i.e., "two" instead of "2."

3. "aposymbiotic" is not hyphenated.

4. Since corals have many different symbionts (bacterial, dinoflagellate, and other), and you are ONLY discussing the Symbiodiniaceae, you should refer to them as "endosymbionts" instead of "symbionts." Similarly, avoid using "algal" but instead use "dinoflagellate" or "Symbiodiniaceae."

5. You only need to mention the full genus name the first time you use it. Afterwards, it can be abbreviated.

6. When providing exact p-values, use the "=" instead of ">" or "<" (which should be reserved for when you want to emphasize whether the value is above or under some preset threshold (e.g. 0.05).

7. Sometimes you abbreviated figure as "Fig" and sometimes as "Fig." Check a recent issue of PLoS ONE to see the preferred nomenclature and maintain consistency throughout.

8. Lines 244-245. Why use a mixed-effect model? Is this because you tracked individuals, in which case there were repeated measures? I would state the justification for this in the text.

9. Lines 211-212: Do you mean that larvae were alive (vs. fixed in formalin)? If so, I would rephrase this for clarity.

10. Line 199: No need to capitalize words in websites.

11. Lines 187-188: Since you mention a complete sentence i parentheses, you need a period (".") prior to the closing parentheses (i.e., ".)").

12. Line 183 and elsewhere: The semicolon should come directly after a word, not after a space. I think this may be because you used the justified setting (it should be the leftwards orientation instead).

13. Sometimes you use "Supplemental S1 dataset" and sometimes "S1 dataset." I think PLoS ONE prefers the latter.

14. Lines 122-123: This statement needs a reference.

15. For Figure 1d (super cool!), would it be possible to put a scale bar on the right of panel? It is hard to know is that part of the image is at the same scale as the flask. It's actually hard to tell what is being shown in general, but I think the figure caption explains it.

16. Figure 7: awesome! I have never heard of a sunflower plot in my life, but I like it.

17. Consider putting the statistical findings (e.g., ANOVA) in a consolidated table (Table 1).

18. Lines 360-362: This should be written in past tense.

19. Line 341: Two periods (only need one).

20. Line 318: 50-ms (since it is being used as an adjective).

All in all, this is a solid article, and I look forward to seeing it published in PLoS ONE very soon. I apologize for the length of time it took to have it peer-reviewed. As you may know, one normally needs to ask 20-30 people to have 2-3 agree. It's incredibly annoying since I feel like there are just a few of us effectively reviewing all the papers!

Anderson

We look forward to receiving your revised manuscript.

Kind regards,

Anderson B. Mayfield, Ph.D.

Academic Editor

PLOS ONE
---

## [Author Response · Author response to Decision Letter 3]

15 Apr 2021

Please see our attached "Responses to Academic editor" file.

---

## [Editor Report · Decision Letter 4]

27 Apr 2021

PONE-D-20-25327R4

Can Acropora tenuis larvae attract native Symbiodiniaceae cells by green fluorescence at the initial establishment of symbiosis?

PLOS ONE

Dear Dr. Yamashita,

    I modified it to "major revision" to where you can resubmit the files. 

We look forward to receiving your revised manuscript.

Kind regards,

Anderson B. Mayfield, Ph.D.

Academic Editor

PLOS ONE

Journal Requirements:

Additional Editor Comments (if provided):

Hello,

I modified it to "major revision" to where you can resubmit the files.

---

## [Author Response · Author response to Decision Letter 4]

29 Apr 2021

We have replaced the S1 Appendix file.

---

## [Editor Report · Decision Letter 5]

18 May 2021

Can Acropora tenuis larvae attract native Symbiodiniaceae cells by green fluorescence at the initial establishment of symbiosis?

PONE-D-20-25327R5

Dear Dr. Yamashita,

We’re pleased to inform you that your manuscript has been judged scientifically suitable for publication and will be formally accepted for publication once it meets all outstanding technical requirements.

Kind regards,

Anderson B. Mayfield, Ph.D.

Academic Editor

PLOS ONE

Additional Editor Comments (optional):

Thank you for bearing with the painstakingly slow peer reviewed process. I thought that during Covid, people would have MORE time to review articles, but I guess they are busy taking care of kids and what not. Anyway, I am happy to say that your manuscript is now ready to be published in PLoS ONE.
---

## [Editor Report · Acceptance letter]

21 May 2021

PONE-D-20-25327R5 

Can *Acropora tenuis* larvae attract native Symbiodiniaceae cells by green fluorescence at the initial establishment of symbiosis? 

Dear Dr. Yamashita:

I'm pleased to inform you that your manuscript has been deemed suitable for publication in PLOS ONE. Congratulations! Your manuscript is now with our production department. 

Kind regards, 

on behalf of

Dr. Anderson B. Mayfield 

Academic Editor

PLOS ONE